# Effects of Compound Biochar Substrate Coupled with Water and Nitrogen on the Growth of Cucumber Plug Seedlings

Guoxin Ma [1,2], Xi Chen [1], Yang Liu [2,3], Jianping Hu [1,2], Luhua Han [1,2] and Hanping Mao [1,2,*]

1   School of Agricultural Engineering, Jiangsu University, Zhenjiang 212013, China
2   Key Laboratory of Modern Agricultural Equipment and Technology, Ministry of Education, Jiangsu University, Zhenjiang 212013, China
3   Machinery Equipment Research Institute, Xinjiang Academy of Agricultural and Reclamation Science, Shihezi 832000, China
*   Correspondence: maohp@ujs.edu.cn; Tel.: +86-1351-169-5868

**Abstract:** Since plug seedlings play a key role in automatic transplanting, this work aimed to explore the interaction between the biochar rate, water content, and N–fertilization in the substrate on the cultivation of cucumber seedlings before and after transplanting. The research showed that most of the factors obtained significant individual and interaction effects by measuring and analyzing the growth parameters of seedlings before transplanting. Most growth parameters significantly decreased with the increase in biochar rate except Water Use Efficiency which obtained the highest value of 2.06 g/L when the biochar rate was 10%. Furthermore, some growth parameters increased significantly with the increase in water content, while the Total Dry Matter and Water Use Efficiency reached their highest values, 0.778 g and 1.94 g/L, respectively, when the water content was 65%. All growth parameters reached their highest values when the N–fertilization was 50%; too high or too low of N–fertilization was not conducive to the growth of seedlings. The growth parameters and photosynthesis indices of seedlings cultivated after transplanting indicated that the seedlings with superior growth before transplanting performed better than other treatments in regard to growth and photosynthesis after transplanting. The interactions were in general optimal when the biochar rate was 5%, water content was 80%, and N–fertilization was 50% in the substrate, and seedlings cultivated under this treatment could not only meet the requirements of automatic transplanting, but also ensure rapid growth after transplanting. This study thus provides some guidance for the effective cultivation of vegetable plug seedlings.

**Keywords:** biochar–water–N interaction; plug seedling; transplanting; cultivation; root system; dry biomass; crushing resistance; photosynthesis

## 1. Introduction

China has the highest vegetable production and consumption in the world now, with a vegetable planting area of 20 million hectares and an annual output of more than 750 million tons [1]. As a key link in the development of the vegetable industry, seedling cultivation and transplanting can effectively improve the yield and quality of vegetables, bringing comprehensive benefits to the vegetable industry [2]. With the decrease in the labor force and the increase in labor cost, automatic seedling–picking technology has developed rapidly in cooperation with the seedling cultivation industry to achieve higher economic benefits [3]. In comparison with manual seedling picking, the automatic seedling picking method puts forward high requirements for the quality of plug seedlings, i.e., the seedling will be easily damaged and the success ratio decreased during the seedling picking process if the mechanical characteristics of the seedling are not good, especially when picking at a high speed [4,5], so it is important to cultivate seedlings with good mechanical properties which can meet the requirements of the high–speed seedling picking technology [6].

Vegetable plug seedlings with strong stems, roots, and substrates are suitable for mechanized automatic seedling picking, and many researchers have made great efforts in cultivating satisfactory seedlings. Yang (1991) proposed that the root system of plug seedlings was the key to binding the loose substrate into a whole and the main stress object during the process of seedling picking [7], therefore, seedlings with strong root systems would possess good mechanical properties. Some research proposed that the mixing of peat, vermiculite, and perlite in different proportions with different nutrient ratios could promote the growth of the root system of cucumber and tomato plug seedlings [8,9] and that the addition of other nutrients to the substrate could also enhance the seedlings. By mixing peat moss and coir as a substrate, Min (2016) found that peat moss could improve the cohesion of the root system and substrate [10]. In addition, Ting (1990) suggested that adding rock wool, with a better fiber structure, into the substrate would improve the binding ability of the root system [11]. Different from these studies, Qu (2018) added a certain proportion of modified urea–formaldehyde resins into the substrate, binding the substrate together in the form of chemical additives [12]; although the strength of the plug seedlings was enhanced to a certain extent by this method, it was not a conventional way to cultivate plug seedlings, and its practicability needs to be verified.

Biochar is a carbon–rich product that can promote the growth of plants by modifying soil conditions. The total porosity of soil can be improved with the addition of biochar, due to its large specific surface area [13], and soil respiration enhanced, which is conducive to crop growth [14]. Relevant research showed that the addition of biochar in the soil affects the dynamics of N in the soil and that biochar can effectively reduce the volatilization of $NH_3^+$ of the soil and convert $NH_3^+$ into $NO_3^-$, which could achieve a slow–release effect of N fertilizer and provide sufficient nutrients for crop and root growth and improve N use efficiency [15–17]. This effect was more obvious when biochar was mixed with N in the soil. Furthermore, the addition of biochar in the soil can increase the water–holding capacity of the soil, which is also beneficial to crop and root growth [18]. It was concluded that adding 10% biochar to soil could maximize the soil water holding capacity by exploring the influence of different biochar amounts on the soil water holding capacity [19]. More importantly, biochar can effectively slow down the impact of soil water stress on crop growth, which could improve root growth, the productivity of crops under water shortage conditions, and water use efficiency [20]. The water and nitrogen content in the culture substrate are also important factors that directly affect the growth of crops, especially the growth of the root system [21]. Research showed that rational water–saving management at the early growth stage of seedlings could enhance the nutrient absorption capacity of seedlings and improve their water use efficiency, which could bring better root distribution [22]. Meanwhile, the reasonable management of nitrogen fertilizer in the substrate could create a strong and dense root system and deeper root distribution, which is conducive to the absorption of water and nutrients by crops [23]. Moreover, the interactive effects of soil water and N management plays a significant role in promoting crop and root growth [24]. Therefore, it is worth exploring the interactive effects of biochar, water, and N on crop growth to discover the most beneficial combination for seedling cultivation.

Based on our previous research, which showed that the addition of 10% biochar in peat could significantly promote the stem, leaf, and root growth of cucumber plug seedlings and enhance their mechanical properties [25], this research refines the proportion of biochar mixed in the substrate and adds different water contents and N–fertilization into the substrate, according to the requirements of seedling cultivation. By measuring the corresponding growth parameters of cucumber seedlings cultivated before and after transplanting, the appropriate biochar rate, water content, and N–fertilization in the substrate for optimal seedling cultivation will be explored, and the coupling relationship between biochar, water, and N fertilizer will also be confirmed. This research could, therefore, provide some basis for the current automatic seedling–picking technology by discovering a suitable cultivation method.

## 2. Materials and Methods

### 2.1. Experimental Site and Materials

The cultivation experiment was conducted from July to September 2022, under a constant temperature culture room and an experimental sunlight greenhouse with controllable light and temperature conditions, at the Key Laboratory of Modern Agricultural Equipment and Technology, Jiangsu University, Zhenjiang, China (32°20′44″ N, 119°45′27″ E). The average monthly temperatures of 30.15 °C in July, 30.54 °C in August, and 23.09 °C in September during the experimental period, and sunshine was sufficient.

The experimental substrate (Supplementary Figure S1a) was mainly composed of peat (K–413, made by Klasmann–Deilmann Co., Geeste, Germany) and biochar. The peat used for plug seedling cultivation had a whole porosity of 62.67%, pH of 5.5~6.0, EC of 0.5 (Dutch Standards), and a total N of 1173.91 mg/100 g. The biochar was obtained by the pyrolysis of pine pellet particles in a tubular reactor at 550 °C [25], with a total N of 495.27 mg/100 g. The soil type selected from Zhenjiang, China, which was used after transplanting, was loam, with a field capacity of 23.1% and a total N of 156 mg/100 g. The peat, biochar, and soil were sieved by passing through a 5 mm mesh before filling the experimental plug tray.

The cucumber seedlings tested in this study were "Jinyou 1", which were provided by the cucumber research institute at the Tianjin Academy of Agricultural Sciences, Tianjin, China. The cucumber seedlings were grown in the incubator (Supplementary Figure S1b) with preset parameters (Supplementary Figure S1c), using the 72–hole plastic plug tray (made by Qihang Co., Suqian, China), before transplanting. The substrate used before transplanting was composed of peat and biochar. After 23 days of growth in the plug tray, the cucumber seedlings were transplanted into the ordinary plastic flowerpots (21 cm in diameter and 12 cm in height, made by Xinglong Plastic Co., Shanghai, China), and cultivated until the end of the flowering period (the overall flowering rate was above 95%). The substrate used after transplanting was composed of peat and soil in a 1:1 ratio, and the nutrient solution prepared according to reference [26] was poured into the substrate every one to two days, according to the weather conditions, in order to ensure full growth of cucumber seedlings.

### 2.2. Experimental Design and Implementation

The experimental treatments included three biochar rates, three water levels, and three N–fertilization amounts. This experimental plan included a total of 27 treatments (i.e., $3 \times 3 \times 3$), and each treatment was replicated twelve times before transplanting, leading to a total of 324 seedlings. In addition, a control group was established with a biochar rate of 0%, a water content of 100%, and an N–fertilization amount of 0% in the substrate to compare with the treatments above.

Based on previous research, the three biochar rates were 5%, 10%, and 15%, which are represented as $B_5$, $B_{10}$, and $B_{15}$, respectively.

Miao (2013) proposed that the cucumber seedling achieved good mechanical properties when the water content of the substrate was about 65% [27], and the growth of plug seedlings would be affected if the water content was too high or too low, so this research chose three water levels, 50%, 65%, and 80%, which are represented as $W_{50}$, $W_{65}$, and $W_{80}$, respectively. During the experimental period, the substrate water content was maintained at the target level by weighing the plug tray and irrigating each hole with tap water by syringe every day.

Advanced Vegetable Physiology indicated that 0.45 g urea should be added to 1 kg substrate to facilitate cucumber growth [28], namely, 0.21 g N should be added to 1 kg substrate during cucumber seedling cultivation. The three N–fertilization amounts were hence set as $N_0$ (0 g N added to the substrate), $N_{50}$ (0.105 g N added to the substrate), and $N_{100}$ (0.21 g N added to the substrate). In addition to N–fertilization, 0.13 g $P_2O_5$, 0.21 g KCl, and 0.018 g $ZnSO_4$ per kg substrate were applied to meet the cucumber seedlings' requirement for successful growth.

*2.3. Sampling and Measurements*

The cucumber plug seedlings of each treatment were cultivated in the incubator for 23 days (from 26 July to 18 August) until they were suitable for transplanting, and eight seedlings were selected from each treatment for testing before transplanting. The remaining four seedlings of each treatment were transplanted into the flowerpot and cultivated until the flowering stage (from 18 August to 5 September) for subsequent tests.

2.3.1. Measurements before Transplanting

1.   Plant height and total root length.

Plant height: The distance from the surface of the substrate to the highest growth point of the seedling as measured by a ruler (0.1 mm) (Supplementary Figure S2a).

Total root length: The root systems of the seedlings in each hole were washed and cleaned with tap water until the substrates were totally removed. Then, the Perfection V700 photo scanner (made by EPSON Co., Nagano, Japan) was used to scan the roots (Supplementary Figure S2b), and the WinRHIZO root analysis software (professional version) was used to process the root data.

2.   Total dry biomass and specific root length.

The seedling substrate was washed away so only the root system remained (Supplementary Figure S2b) and absorbent paper was used to absorb excess water. The seeding was then put into an oven set at 105 °C for 15 min, after which the temperature was reset to 80 °C and the seedling dried until the weight remained constant; finally, a precision electronic scale (0.0001 g) was used to obtain the weight of the root dry matter.

In addition, the total dry biomass (Total *DM*) and specific root length (*SRL*) of the seedlings were calculated using the following equations:

$$\text{Total } DM = \text{Shoot } DM + \text{Root } DM, \tag{1}$$

$$SRL = \text{Total root length}/\text{Root } DM, \tag{2}$$

3.   Water and N use efficiency.

Water use efficiency (*WUE*): Since the maximum water content of the substrate was 80% in the experiment, no leakage occurred during the plant growth, so all water was absorbed by seedlings except evaporated water. In order to ensure the accuracy of the *WUE*, a standard evaporating dish was put into the incubator during seedling cultivation to obtain the evaporation amount of water in the incubator environment; therefore, *WUE* was calculated as:

$$WUE = \text{Shoot } DM/(TIW - SW - TEW), \tag{3}$$

where *TIW* is total irrigation water during plant growth, L; *SW* is the actual substrate weight on the 23rd day minus the initial substrate weight, and converted into volume, L; and *TEW* is the total amount of water evaporated, L.

N use efficiency (*NUE*): All dried samples were ground and sifted through a 0.5 mm mesh to analyze the total N concentration using a UDK 159 automatic Kjeldahl azotometer (made by VELP Co., Milan, Italy). First, N absorption in shoots (*NA*) was calculated as:

$$NA = \text{Shoot } DM \times NCS, \tag{4}$$

where *NCS* is N content in the shoot of seedlings, %.
Then, *NUE* was calculated as:

$$NUE = (NAL - NAI)/NSI \times 100\%, \tag{5}$$

where *NAL* is the N uptake amount of shoot cultivated after adding N in the substrate, g; *NAI* is the N uptake amount of shoot cultivated without adding N in Substrate, g; and *NSI* is N content in the substrate, g.

4. Crushing resistance of the seedling pot.

Free dropping tests of seedlings, cultivated for 23 days, were conducted in each treatment in order to verify the mechanical properties of these seedling pots. The seedlings were picked out from the tray and weighed, dropped freely from a height of 0.5 m to the ground in a vertical state, and the damaged seedling pots were weighed after dropping. Each treatment was replicated three times, and the values were averaged. Thus, the crushing resistance of the seedling pot (*RC*) was calculated as:

$$RC = WD/WI \times 100\%,$$ (6)

where *WD* is the weight of damaged seedlings after dropping, g and *WI* is the weight of initial seedlings before dropping, g.

In order to verify the accuracy of the optimal parameters promoting the growth of cucumber seedlings, the extra seedling cultivation experiment was carried out according to the optimal factor combination obtained above and a similar combination and worst combination were arranged for comparative experiments; then, the crushing resistance was finally analyzed. Three groups of experiments were designed for each combination, and a whole tray of 72 cucumber seedlings was cultivated for each group to guarantee the accuracy of the experiment.

2.3.2. Measurements after Transplanting

1. Plant height, shoot dry biomass, total root length, and root dry biomass.

The plant height of the cucumber seedlings was measured by a steel tape (0.1 mm) according to the method above when the transplanted seedlings grew to the flowering stage (Supplementary Figure S3b). After the flowering stage, the seedlings were harvested. The roots were washed carefully with tap water following the method above, and the total root length was measured by a Perfection V700 photo scanner; then, the shoots and roots were put into an oven, dried, and weighed for further analysis.

2. The leaf gas exchange parameters.

The leaf gas exchange parameter is a major reflection of the photosynthesis of the seedlings and an important parameter to gauge their physiological indexes. In this study, the LI–6400 portable photosynthesis measuring system (made by LI–COR Co., Superior Street Lincoln, NE, USA) was selected to measure the net photosynthetic rate, transpiration rate, intercellular $CO_2$ concentration, and stomatal conductance of the seedling leaves (Supplementary Figure S3c). In order to ensure the accuracy of the test data, the photosynthetic data measurement was conducted at 9:00~11:00 in the morning on a sunny day, and the position of the selected leaves on each seedling was basically the same.

*2.4. Statistical Analyses*

All experimental data were analyzed by using SPSS 23.0 and Excel 2019, and all data graphs were made by Origin 2018. The complete random analysis of variance (ANOVA) method was used for multiple comparisons of these data, and the means were compared using the least significant difference (LSD) test at a probability level of 0.05.

**3. Results**
*3.1. Data Analysis of Seedlings before Transplanting*
3.1.1. Analysis of Plant Height and Total Root Length

The measurements of the plant height and total root length of the cucumber seedlings under different treatments are shown in Table 1.

**Table 1.** Plant height, total root length, total dry biomass, and specific root length of cucumber seedlings.

| Factors and Levels | Plant Height /mm | Total Root Length /cm | Total Dry Biomass /g | Specific Root Length /cm·mg$^{-1}$ |
|---|---|---|---|---|
| Biochar level (B) | | | | |
| $B_5$ | 58.4 a | 116.17 a | 0.0802 a | 23.83 a |
| $B_{10}$ | 57.6 a | 113.32 a | 0.0778 b | 22.94 a |
| $B_{15}$ | 52.2 b | 99.39 b | 0.0643 c | 15.27 b |
| Water level (W) | | | | |
| $W_{50}$ | 53.9 b | 95.10 c | 0.0674 b | 15.08 c |
| $W_{65}$ | 56.4 a | 110.72 b | 0.0778 a | 21.00 b |
| $W_{80}$ | 57.1 a | 125.07 a | 0.0749 a | 25.97 a |
| Nitrogen level (N) | | | | |
| $N_0$ | 55.7 b | 109.49 b | 0.0763 a | 20.54 b |
| $N_{50}$ | 57.1 a | 122.13 a | 0.0737 a | 24.44 a |
| $N_{100}$ | 54.4 b | 97.26 c | 0.0732 a | 17.06 c |
| Analysis of variance | | | | |
| B | *** | ** | *** | *** |
| W | ** | *** | ** | *** |
| N | * | *** | ns | *** |
| B × W | * | *** | ** | * |
| B × N | *** | *** | ** | *** |
| W × N | ** | * | * | ** |
| B × W × N | *** | ** | ** | *** |

Note: Values with different lower case letters were used in the same column to indicate the significance of the difference when $p < 0.05$; *** ($p < 0.001$), ** ($p < 0.01$), * ($0.01 < p < 0.05$), and ns ($p > 0.05$) were used to indicate whether there was remarkable difference between different treatments. The same as below.

The biochar rate, water content, and N–fertilization in the substrate showed a significant individual effect on plant height. Compared with other treatments, the $B_5$ treatment obtained the best plant height while the $B_{10}$ treatment was slightly lower than that of the $B_5$ treatment, and there was no significant difference between $B_5$ and $B_{10}$ treatments. However, the plant height of the $B_{15}$ treatment was much lower than the $B_5$ and $B_{10}$ treatments, showing a significant comparative difference. This result indicated that 5% biochar mixed into the substrate could significantly promote the growth of cucumber seedlings, but a higher biochar rate was not actually better. The water content in the substrate also showed a significant individual effect on plant height. The best plant height was obtained in the $W_{80}$ treatment; the plant height with the $W_{65}$ treatment was slightly lower than that of the $W_{80}$ treatment, and the difference was not significant. However, the plant height of the $W_{50}$ treatment was significantly lower than that of the other two treatments because the cucumber seedlings were in a state of water shortage under the $W_{50}$ treatment, and the water content in the substrate could not guarantee the normal growth of cucumber seedlings. Therefore, the water content most conducive to the growth of cucumber seedlings was 80% or 65%. The plant height under the $N_{50}$ treatment was higher than those of other treatments. While the $N_0$ and $N_{100}$ treatments were lower than that of the $N_{50}$ treatment, the difference in plant height between the $N_0$ and $N_{100}$ treatments was not significant. This result indicated that the initial N present in the substrate could meet the growth of cucumber seedlings to some extent but adding a certain proportion of N (50%) could significantly improve the growth trend of the cucumber seedlings. However, the growth of cucumber seedlings would be inhibited if the proportion of N added was too high (100%). Therefore, the amount of N fertilizer added to the substrate must be appropriate.

The biochar rate, water content, and N–fertilization in the substrate also showed a significant individual effect on total root length. The total root length of cucumber seedlings under the $B_5$ treatment was longer than those of the other treatments; it was slightly shorter under the $B_{10}$ treatment, but it was the shortest under the $B_{15}$ treatment. There were

also significant differences between the two values obtained from different water content treatments, and the total root length order of the cucumber seedlings was $W_{80} > W_{65} > W_{50}$. The total root length of the cucumber seedlings under the $N_{50}$ treatment was longer than those of other treatments, and the value of the $N_0$ treatment was lower than that of the $N_{50}$ treatment but higher than that of the $N_{100}$ treatment.

Total root length was an important factor affecting the mechanical properties of the cucumber seedlings [10], and the biochar rate, water content, and N–fertilization in the substrate had obvious two–way interactions with the total root length. The interactions between the factors on the total root length are shown in Figure 1.

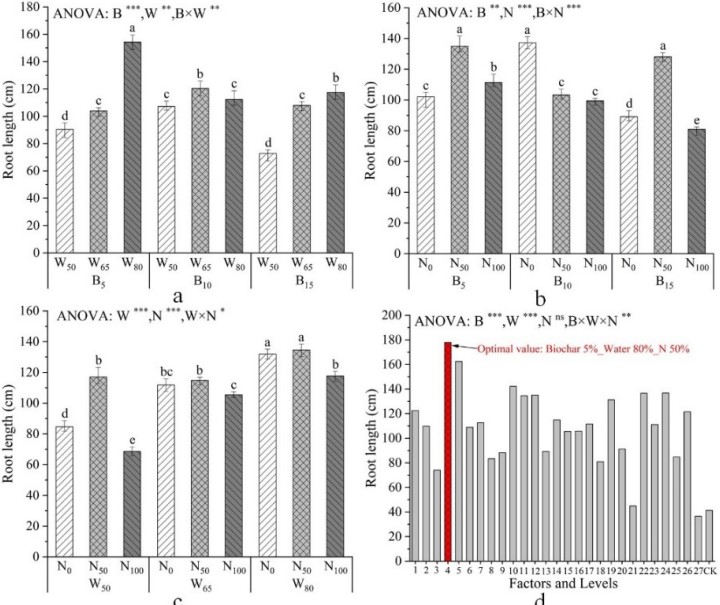

**Figure 1.** Effects of the total root length with different treatments of seedlings: (**a**) Interaction between biochar and water content; (**b**) Interaction between biochar and N; (**c**) Interaction between water content and N; and (**d**) Interaction between biochar, water, and N. *Note:* The implication of the 27 numbers and CK on the *x*–axis in the Figure is explained in Supplementary Table S1; values with different lower case letters were used in the same picture to indicate the significance of the difference when $p < 0.05$; *** ($p < 0.001$), ** ($p < 0.01$), * ($0.01 < p < 0.05$) were used to indicate whether there was remarkable difference between different treatments. The same as below.

As shown in Figure 1a, the interaction between biochar and water on total root length was comparatively significant. The total root length under the $B_5$—$W_{80}$ treatment was the longest, while the total root length under the $B_{15}$–$W_{50}$ treatment was the shortest. Under the treatments with $B_5$ and $B_{15}$, the total root length increased significantly with the increase in water content, reaching the maximum value under the $W_{80}$ treatment. However, under the treatment of $B_{10}$, the $W_{65}$ treatment had the longest total root length, while the $W_{50}$ treatment had the shortest total root length.

As shown in Figure 1b, the interaction between biochar and N on total root length was extremely significant. The total root length under the $B_5$–$N_{50}$ treatment was the longest, while it was the lowest under the $B_{15}$–$N_{100}$ treatment. The response of total root length to N–fertilization was also different under different biochar rates. Under the treatments of $B_5$ and $B_{15}$, the total root length reached the maximum under the $N_{50}$ treatment, but the total root length decreased significantly with the increase in N–fertilization under the $B_{10}$ treatment.

As shown in Figure 1c, the interaction between water content and N on total root length was significant. The total root length under the $W_{80}$–$N_{50}$ treatment was the longest, while the $W_{50}$–$N_{100}$ treatment had the shortest total root length. The response trend of the total root length to N content was relatively consistent under different water contents, and

the total root length was always the longest in the $N_{50}$ treatment under all water contents; the total root length order was always $N_{50} > N_0 > N_{100}$.

Figure 1d indicates that the biochar rate, water content, and N–fertilization had significant interactions on the total root length, and the final results of the interaction between three factors showed that the $B_5$–$W_{80}$–$N_{50}$ treatment obtained the longest total root length which was superior to those of other treatments and far greater than that of the CK treatment.

### 3.1.2. Analysis of Total Dry Biomass and Specific Root Length

The results of the total dry biomass and specific root length of cucumber seedlings under different treatments are also shown in Table 1.

The biochar rate had significant individual effects on the total dry biomass and specific root length. The total dry biomass and specific root length of cucumber seedlings were better than those of other treatments under the $B_5$ treatment and showed a downward trend with the increase in the biochar rate. The water content also had significant individual effects on the total dry biomass and specific root length. The two parameters decreased with the decrease in water content, and the optimal value was obtained under the $W_{80}$ treatment, while the decrease was particularly obvious under the $W_{50}$ treatment. In addition, N–fertilization had no significant effect on the total dry biomass, but only had an individual significant effect on specific root length. In general, the total dry biomass and specific root length of cucumber seedlings under the $N_{50}$ treatment were better than those under other treatments, while the values obtained under the $N_{100}$ treatment were the lowest. This trend was more obvious in specific root lengths.

Concerning the ratio of root length to root dry biomass, the specific root length could represent a certain relationship between the root growth parameters and dry biomass, i.e., the lower the value, the larger the average diameter of the root system [29]. The interactions among three factors on specific root length were analyzed and are shown in Figure 2,.

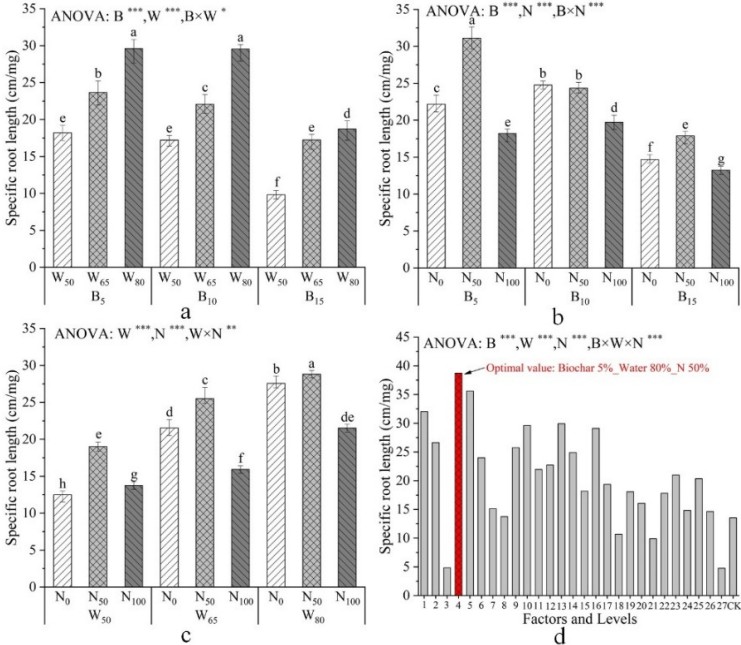

**Figure 2.** Effects of the specific root length with different treatments of seedlings: (**a**) Interaction between biochar and water content; (**b**) Interaction between biochar and N; (**c**) Interaction between water content and N; and (**d**) Interaction between biochar, water content, and N. Note: Values with different lower case letters were used in the same column to indicate the significance of the difference when $p < 0.05$; *** ($p < 0.001$), ** ($p < 0.01$), * ($0.01 < p < 0.05$) were used to indicate whether there was remarkable difference between different treatments.

As shown in Figure 2a, the interaction between biochar and water content on specific root lengths was significant. The specific root length was the largest under the $B_5$–$W_{80}$ treatment, while that under the $B_{15}$–$W_{50}$ treatment was the smallest. The response of the specific root length to water content was consistent under different biochar rates; the specific root length increased with the increase in water content and reached the highest value in the $W_{80}$ treatment under all biochar rates.

As shown in Figure 2b, the interaction between biochar and N on specific root length was extremely significant. The specific root length was the largest under the $B_5$–$N_{50}$ treatment and the lowest under the $B_{15}$–$N_{100}$ treatment. The response of specific root length to N–fertilization was similar under that of different biochar rates, and the maximum value was reached in the $N_{50}$ treatment under the $B_5$ and $B_{15}$ treatments, which was significantly different from those of other treatments. In addition, the difference between the $N_0$ and $N_{50}$ treatments was not significant under the $B_{10}$ treatment, but the value of the $N_{100}$ treatment was the lowest under all treatments.

As shown in Figure 2c, the interaction between water content and N on specific root lengths was significant. The specific root length was the largest under the $W_{80}$–$N_{50}$ treatment and the smallest under the $W_{50}$–$N_0$ treatment, and the response trend of the specific root length to N–fertilization was consistent under different water contents. The specific root length first increased and then decreased with the increase in N–fertilization, and finally reached the maximum value in the $N_{50}$ treatment; there was a significant difference between the different treatments.

Figure 2d indicates that the biochar rate, water content, and N–fertilization had extremely significant interactions on the specific root length, and the final results of the interaction between the three factors show that the $B_5$–$W_{80}$–$N_{50}$ treatment obtained the largest specific root length, which was slightly larger than that of $B_5$–$W_{65}$–$N_{50}$ treatment.

### 3.1.3. Analysis of Water and N Use Efficiency

The results of the water use efficiency and N use efficiency of cucumber seedlings under different treatments are shown in Table 2.

**Table 2.** N use efficiency, water use efficiency, and crushing resistance of cucumber seedlings.

| Factors and Levels | N use Efficiency /% | Water Use Efficiency /g·L$^{-1}$ | Crushing Resistance /% |
|---|---|---|---|
| Biochar level (B) | | | |
| $B_5$ | 32.03 a | 1.56 b | 84.59 a |
| $B_{10}$ | 30.35 a | 2.06 a | 80.99 b |
| $B_{15}$ | 26.93 b | 1.43 b | 76.63 c |
| Water level (W) | | | |
| $W_{50}$ | 26.42 c | 1.61 b | 74.55 b |
| $W_{65}$ | 29.84 b | 1.94 a | 83.81 a |
| $W_{80}$ | 33.06 a | 1.50 b | 84.84 a |
| Nitrogen level (N) | | | |
| $N_0$ | — [n] | 1.44 c | 78.37 c |
| $N_{50}$ | 32.32 a | 1.96 a | 83.30 a |
| $N_{100}$ | 27.23 b | 1.66 b | 80.54 b |
| Analysis of variance | | | |
| B | ** | *** | ** |
| W | ** | *** | *** |
| N | *** | *** | ** |
| B×W | *** | *** | *** |
| B×N | *** | *** | ** |
| W×N | * | ** | *** |
| B×W×N | ns | *** | *** |

[n] This column is empty and has no corresponding value. Note: Values with different lower case letters were used in the same column to indicate the significance of the difference when $p < 0.05$; *** ($p < 0.001$), ** ($p < 0.01$), * ($0.01 < p < 0.05$), and ns ($p > 0.05$) were used to indicate whether there was remarkable difference between different treatments.

Biochar rate, water content, and N–fertilization had significant individual and interactive effects on water use efficiency and N use efficiency, but the interaction among the three factors in N use efficiency was not significant. The N use efficiency of cucumber seedlings under the $B_5$ treatment was higher than those of other treatments, while the water use efficiency reached the optimal value under the $B_{10}$ treatment, but the two parameters both obtained the lowest value under the $B_{15}$ treatment. The N use efficiency was the highest under the $W_{80}$ treatment, while the water use efficiency reached the optimal value under the $W_{65}$ treatment. In comparison, the N and water use efficiency under the $W_{50}$ treatment had the lowest value. Moreover, the N and water use efficiency reached the highest value under the $N_{50}$ treatment, and the difference between different treatments was significant.

According to the calculation rules, it was impossible to definitively calculate the N–use efficiency of cucumber seedlings under the treatment of $N_0$. Therefore, we selected the water use efficiency to analyze the interaction among three factors, and the results are shown in Figure 3.

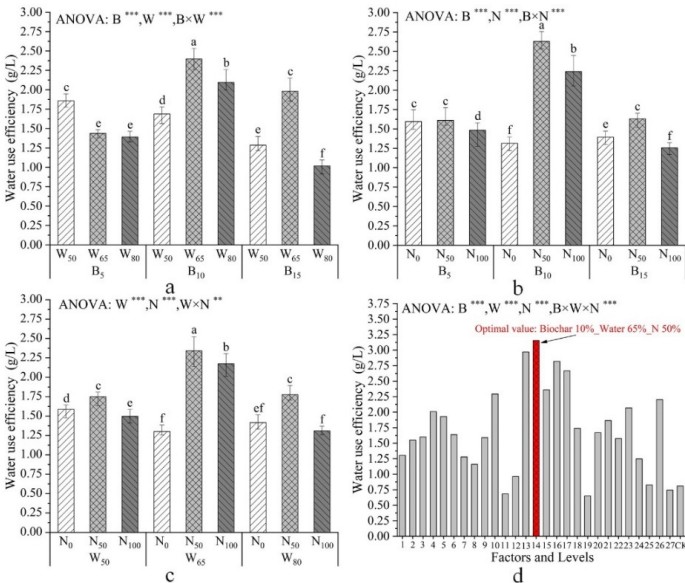

**Figure 3.** Effects of water use efficiency with different treatments of seedlings: (**a**) Interaction between biochar and water content; (**b**) Interaction between biochar and N; (**c**) Interaction between water content and N; and (**d**) Interaction between biochar, water content, and N. Note: Values with different lower case letters were used in the same column to indicate the significance of the difference when $p < 0.05$; *** ($p < 0.001$), ** ($p < 0.01$) were used to indicate whether there was remarkable difference between different treatments.

As shown in Figure 3a, the interaction between biochar and water content on water use efficiency was extremely significant. The water use efficiency obtained the highest value under the $B_{10}$–$W_{65}$ treatment, while that of the $B_{15}$–$W_{80}$ treatment had the lowest value. The response of water use efficiency to water content was inconsistent under different biochar rates; the water use efficiency of the $W_{50}$ treatment was higher than those of other treatments under the $B_5$ treatment, while the water use efficiency was the highest in the $W_{65}$ treatment under the $B_{10}$ and $B_{15}$ treatments.

As shown in Figure 3b, the interaction between biochar and N on water use efficiency was extremely significant. The water use efficiency was the highest under the $B_{10}$–$N_{50}$ treatment and the lowest under the $B_{15}$–$N_{100}$ treatment. The response of water use efficiency to N content was quite different under different biochar rates, but the water use efficiency was the highest in the $N_{50}$ treatment under various biochar rates, and the difference in water use efficiency with different N–fertilizations was the most significant under the $B_{10}$ treatment.

As shown in Figure 3c, the interaction between water content and N on water use efficiency was comparatively significant. The water use efficiency obtained the highest value under the $W_{65}$–$N_{50}$ treatment, while the lowest was under the $W_{80}$–$N_{100}$ treatment, and the response of water use efficiency to N content was basically the same under different water contents. The water use efficiency first increased and then decreased with the increase in N content, and the highest water use efficiency was obtained in $N_{50}$ treatment under different water contents.

Figure 3d indicates that the biochar rate, water content, and N–fertilization had an extremely significant interaction on the water use efficiency, and the final results of the interaction among the three factors show that the $B_{10}$–$W_{65}$–$N_{50}$ treatment obtained the highest water use efficiency.

### 3.1.4. Analysis of Crushing Resistance of Seedling Pot

The results of the crushing resistance of seedling pots under different treatments are shown in Table 2.

Biochar rate, water content, and N–fertilization had significant individual effects on the crushing resistance of the seedling pot. The crushing resistance of seedling pots cultivated in the $B_5$ treatment was better than those of other treatments, and the crushing resistance decreased significantly with the increase in the biochar rate. With the increase in water content, the crushing resistance increased, and the highest value was obtained under the $W_{80}$ treatment, while the $W_{50}$ treatment value was the lowest. The difference was significant in the $W_{80}$ treatment compared with those of other treatments, but the difference between the $W_{65}$ and $W_{50}$ treatments was not significant.

Biochar rate, water content, and N–fertilization had significant interactive effects on crushing resistance; the interaction between the three factors regarding crushing resistance is shown in Figure 4.

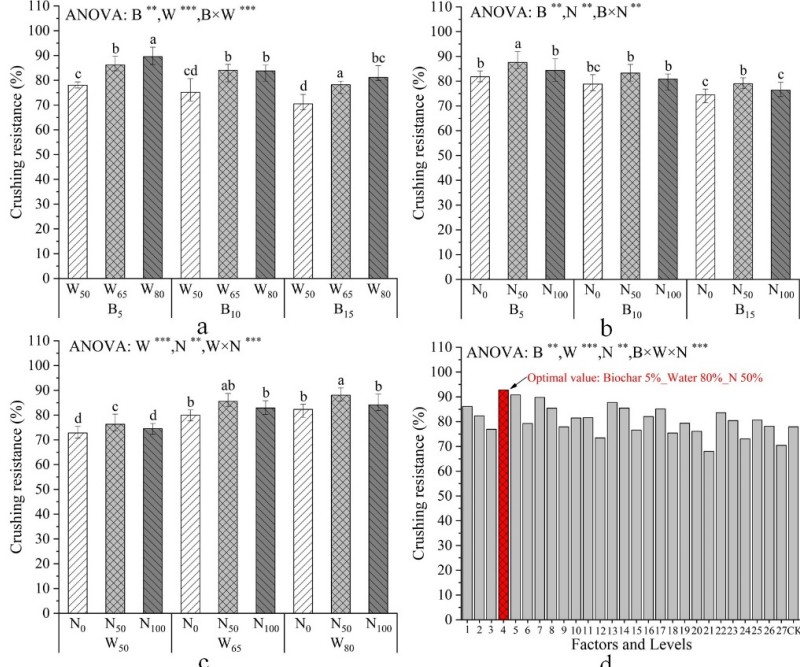

**Figure 4.** Effects of crushing resistance of seedling pot with different treatments of seedlings: (**a**) Interaction between biochar and water content; (**b**) Interaction between biochar and N; (**c**) Interaction between water content and N; and (**d**) Interaction between biochar, water content, and N. Note: Values with different lower case letters were used in the same column to indicate the significance of the difference when $p < 0.05$; *** ($p < 0.001$), ** ($p < 0.01$) were used to indicate whether there was remarkable difference between different treatments.

As shown in Figure 4a, the interaction between biochar and water content on water use efficiency was extremely significant. The seedling pots cultivated under the $B_5$–$W_{80}$ treatment obtained the best crushing resistance, while the $B_{15}$–$W_{50}$ treatment had the worst crushing resistance. The response of crushing resistance to water content was consistent under different biochar rates, and the crushing resistance was significantly improved with the increase in water content, with the highest value obtained under the $W_{80}$ treatment.

As shown in Figure 4b, the interaction between biochar and water content on water use efficiency was comparatively significant. The crushing resistance was the best under the $B_5$–$N_{50}$ treatment and the worst under the $B_{15}$–$N_0$ treatment. The response of crushing resistance to N–fertilization was also consistent under different biochar rates, i.e., the crushing resistance first increased and then decreased with the increase in N–fertilization. The best value of crushing resistance was achieved by the seedlings cultivated in the $N_{50}$ treatment under different biochar rates, while the worst was in the $N_0$ treatment.

As shown in Figure 4c, the interaction between biochar and water content on water use efficiency was extremely significant. The crushing resistance was the best under the $W_{50}$–$N_{50}$ treatment and the worst under the $W_{50}$–$N_0$ treatment. The response of crushing resistance to water content was also consistent under different biochar rates, i.e., the crushing resistance first increased and then decreased with the increase in water content, and the best value of crushing resistance was always obtained from the seedlings cultivated under $N_{50}$ treatment.

Figure 4d indicates that the biochar rate, water content, and N–fertilization had an extremely significant interaction on crushing resistance, and the final results of the interaction among the three factors show that the $B_5$–$W_{80}$–$N_{50}$ treatment obtained the best crushing resistance.

An extra seedling cultivation experiment was carried out under the optimal treatment ($B_5$–$W_{80}$–$N_{50}$), while a similar treatment ($B_5$–$W_{65}$–$N_{50}$) and the worst treatment ($B_{15}$–$W_{50}$–$N_{100}$) were also arranged for comparative experiments. The experimental results are shown in Table 3. The standard deviations of the crushing resistance of the three treatments were 0.9582, 0.9238, and 0.9417, respectively, which indicated that the test results were accurate and effective.

**Table 3.** Verification of the crushing resistance of the cucumber seedlings.

| Combination of Tests | Average Value /% | Maximum Value /% | Minimum Value /% | Standard Deviation |
|---|---|---|---|---|
| $B_5$–$W_{80}$–$N_{50}$ | 91.01 | 92.78 | 90.56 | 0.9582 |
| $B_5$–$W_{65}$–$N_{50}$ | 89.38 | 90.41 | 88.15 | 0.9238 |
| $B_{15}$–$W_{50}$–$N_{100}$ | 73.54 | 75.03 | 72.76 | 0.9417 |

The experimental results showed that the crushing resistance of seedling pots cultivated under the $B_5$–$W_{80}$–$N_{50}$ treatment was still the best, with an average value of 91.01%, and the relative deviation from the results of the previous experiment was only 0.19%. The crushing resistance of seedling pots cultivated under the $B_5$–$W_{65}$–$N_{50}$ treatment was 89.39%, which was basically consistent with previous test results, and only 1.63% lower than that of the $B_5$–$W_{80}$–$N_{50}$ treatment. However, the crushing resistance of seedling pots cultivated under the $B_{15}$–$W_{50}$–$N_{100}$ treatment was only 73.54%, which was far lower than those of the other two treatments. The results verified the correctness of the previous test results and also showed that the optimal treatment ($B_5$–$W_{80}$–$N_{50}$) could cultivate cucumber seedlings with strong crushing resistance.

### 3.2. Data Analysis of Seedlings after Transplanting

3.2.1. Analysis of Plant Height, Shoot Dry Biomass, Total Root Length, and Root Dry Biomass

The experimental results of plant height, shoot dry biomass, total root length, and root dry biomass are shown in Table 4. In general, different treatments on the substrate had

a significant impact on the growth of cucumber seedlings before and after transplanting, and the cucumber seedlings that grew well before transplanting still had the best growth after transplanting.

**Table 4.** The plant height, shoot dry biomass, total root length, and root dry biomass of cucumber seedlings.

| Factors and Levels | Plant Height /mm | Shoot DM /g | Total Root Length /cm | Root DM /g |
|---|---|---|---|---|
| Biochar level (B) | | | | |
| $B_5$ | 234.18 a | 2.033 a | 1150.04 a | 0.108 a |
| $B_{10}$ | 218.52 b | 1.963 a | 1166.67 a | 0.104 a |
| $B_{15}$ | 198.59 c | 1.937 b | 982.85 b | 0.077 b |
| Water level (W) | | | | |
| $W_{50}$ | 191.32 c | 1.551 b | 871.47 c | 0.074 b |
| $W_{65}$ | 217.32 b | 1.837 a | 1157.97 b | 0.096 a |
| $W_{80}$ | 238.64 a | 1.885 a | 1278.13 a | 0.099 a |
| Nitrogen level (N) | | | | |
| $N_0$ | 211.91 b | 1.657 b | 1103.29 b | 0.082 b |
| $N_{50}$ | 228.39 a | 1.952 a | 1256.99 a | 0.104 a |
| $N_{100}$ | 194.99 b | 1.625 b | 901.28 c | 0.101 a |
| $B_0-W_{100}-N_0$ | | | | |
| CK | 140.71 | 1.210 | 652.56 | 0.061 |

Note: Values with different lower case letters were used in the same column to indicate the significance of the difference when $p < 0.05$.

With the increase in the biochar rate before transplanting, the plant height, shoot dry biomass, and root dry biomass of seedlings after transplanting showed a significant downward trend, but the total root length increased first and then decreased significantly. In addition, with the increase in water content before transplanting, the plant height, shoot dry biomass, total root length, and root dry biomass of seedlings after transplanting all showed a significant upward trend. However, the plant height, shoot dry biomass, total root length, and root dry biomass of seedlings showed a trend of first significant increase and then significant decrease with the increase in N–fertilization before transplanting.

The growth trend of seedlings after transplanting was consistent with that before transplanting, but the gap between the maximum and minimum extreme values of each parameter after transplanting increased sharply compared with that before transplanting. Taking the total root length as an example, the gap between the maximum total root length under the $W_{80}$ treatment and the minimum total root length under the $W_{50}$ treatment before transplanting was 23.96%, but the gap became 31.82% after transplanting, increasing by 7.86%. Similarly, the gap between the maximum value of the $N_{50}$ treatment and the minimum value of the $N_{100}$ treatment before transplanting was 20.36%, but it became 28.3% after transplanting, increasing by 7.94%. However, the gap in total root length before and after transplanting was not obvious under different biochar rates, and the gap between the maximum value of the $B_5$ treatment and the minimum value of the $B_{15}$ treatment was 14.44% before transplanting, while the longest total root length after transplanting was under the $B_{10}$ treatment, so the gap between the $B_{10}$ and $B_{15}$ treatments after transplanting became 15.76%, increasing only by 1.32%.

The results above show that the cucumber seedlings with superior growth before transplanting would grow more vigorously after transplanting, while the cucumber seedlings with poor growth before transplanting would grow more slowly after transplanting. Therefore, it is particularly important to cultivate cucumber seedlings with good growth and superior physiological indicators before transplanting.

### 3.2.2. Analysis of the Leaf Gas Exchange Parameters

The leaf gas exchange parameters of the cucumber seedlings cultivated in each treatment after transplanting were measured, and the results are shown in Table 5. The difference in biochar rate, water content, and N–fertilization in the substrate before transplanting significantly affected the photosynthesis of the seedlings' leaves after transplanting.

**Table 5.** Leaf gas exchange parameters of cucumber seedlings.

| Factors and Levels | Net Photosynthetic Rate /$\mu$mol($CO_2$)m$^{-2}$s$^{-1}$ | Transpiration Rate /mmol($H_2O$)m$^{-2}$s$^{-1}$ | Intercellular $CO_2$ Concentration /$\mu$mol($CO_2$)mol$^{-1}$ | Stomatal Conductance /mmol($H_2O$)m$^{-2}$s$^{-1}$ |
|---|---|---|---|---|
| | | Biochar level (B) | | |
| $B_5$ | 14.55 a | 0.88 a | 195.05 a | 10.20 a |
| $B_{10}$ | 14.44 a | 0.90 a | 195.20 a | 10.15 a |
| $B_{15}$ | 14.01 b | 0.84 b | 188.03 b | 9.63 b |
| | | Water level (W) | | |
| $W_{50}$ | 13.51 b | 0.77 b | 192.99 b | 9.69 b |
| $W_{65}$ | 14.26 a | 0.88 a | 191.37 b | 10.02 a |
| $W_{80}$ | 14.53 a | 0.90 a | 197.92 a | 10.17 a |
| | | Nitrogen level (N) | | |
| $N_0$ | 14.28 a | 0.88 a | 192.70 a | 9.64 b |
| $N_{50}$ | 14.41 a | 0.89 a | 193.06 a | 10.16 a |
| $N_{100}$ | 13.83 b | 0.81 b | 185.51 b | 9.49 b |
| | | $B_0$–$W_{100}$–$N_0$ | | |
| CK | 13.16 | 0.75 | 176.83 | 9.13 |

Note: Values with different lower case letters were used in the same column to indicate the significance of the difference when $p < 0.05$.

The net photosynthetic rate, transpiration rate, intercellular $CO_2$ concentration, and stomatal conductance of the seedlings' leaves after transplanting decreased with the increase in the biochar rate; the values of each parameter under the $B_{15}$ treatment were the lowest, while the difference between the $B_5$ and $B_{10}$ treatments was not obvious.

The net photosynthetic rate, transpiration rate, and stomatal conductance of the seedlings' leaves after transplanting increased significantly with the increase in water content, but the intercellular $CO_2$ concentration of the leaves decreased first and then increased with the increase in water content.

The net photosynthetic rate, transpiration rate, and stomatal conductance of the seedlings' leaves after transplanting all increased first and then decreased significantly with the increase in N–fertilization, and each parameter was the lowest under the $N_{100}$ treatment.

### 4. Discussion

Previous research indicated that the management of water content–N content in the soil, biochar–N in soil, and biochar–water content in the soil during crop cultivation had very obvious individual and interactive effects on crop growth [30,31], but it was unclear whether there were individual and interaction effects of the biochar rate, water content, and N–fertilization on cucumber seedling growth. At present, there are excessive watering and nutrient solution applications in cucumber seedling cultivation. Excessive watering can cause damage to the seedling pot and a loss of nutrients, including a waste of water resources, and excessive N–fertilization can lead to excessive nutrition of plug seedlings and the salinization of the substrate. This research was conducive to an in–depth understanding of the complex interaction of biochar, water content, and N on the growth of cucumber seedlings, and also has a certain guiding significance for the industrialized cultivation of plug seedlings.

All detected factors regarding N–fertilization in the substrate had a significant effect on the total dry biomass; there were significant individual and interaction effects on the plant height, total root length, and specific root length of cucumber seedlings under different biochar rates, water contents, and N–fertilizations, and the optimal values were all obtained under the $B_5$–$W_{80}$–$N_{50}$ treatment. Firstly, the addition of biochar in the substrate should be controlled within the range of 5% to 10% to be conducive to the growth of cucumber seedlings. Since drought stress significantly prevents the growth of cucumber seedlings and roots [32], the growth parameters cultivated with 50% water content were much lower than those of other treatments, but this situation could be alleviated with the addition of biochar in the substrate [33]. Compared with drought stress, the higher water content with the addition of biochar in the substrate could obtain a better total root length and specific root length because the biochar and water interaction could promote the growth and wide distribution of roots [34,35]. The cucumber seedlings had the optimum growth parameters under the 50% N treatment; a too high and too low N content affects the seedling growth, especially the root growth, since a high N significantly inhibits the growth of roots and the accumulation of dry biomass [36]. However, the application of a certain proportion of biochar in the substrate with high water content could alleviate the inhibition of high N on seedling growth [37,38].

The N use efficiency of cucumber seedlings also reached the maximum value under the $B_5$–$W_{80}$–$N_{50}$ treatment. This is because a high water content in addition to the application of biochar in the substrate can obtain better root distribution, which can improve the N use efficiency of seedlings [39,40]. However, the N use efficiency of seedlings cultivated under the 100% N treatment obtained the lowest value, as excessive N–fertilization leads to a significant decline in the dry biomass of cucumber seedlings [41]. Since biochar can absorb a large number of nutrients with a large specific surface area, which could partially achieve the slow–release effect of N fertilizer [42], the combined application of biochar and N could significantly improve the N use efficiency of cucumber seedlings. The three factors of the water use efficiency of cucumber seedlings also showed obvious individual and interaction effects, but the highest value was obtained under the $B_{10}$–$W_{65}$–$N_{50}$ treatment. The water use efficiency of cucumber seedlings was the highest when the water content was 65%, but the lowest when the water content was 80%, even lower than the drought stress treatment. The reason is that the water was fully utilized with the decrease in seedling transpiration [43], and the cucumber seedlings could not absorb the whole 80% water, which led to lower water use efficiency, so it was better to improve the water use efficiency with a water content of 65%. Results showed that 10% biochar mixed in the substrate could achieve the highest water use efficiency since the water holding capacity of 10% biochar was more prominent [19,25]. In addition, the application of 100% N in the substrate resulted in the lowest water use efficiency because excessive N reduces the leaf area of seedlings, which leads to a poor leaf assimilation capacity, so the water use efficiency is significantly reduced [44]. Therefore, a 65% water content and 50% N–fertilization in the substrate brings better water use efficiency with a proper biochar rate, and a certain proportion of biochar in the substrate would significantly improve the N and water use efficiency of cucumber seedlings [45].

As shown in Figure 5, when the cucumber seedlings fell to the ground, the damage probability of the parts without root growth was high and the damage was relatively serious. Therefore, the mechanical properties of the seedling pots would be stronger with more developed root systems in cucumber seedlings. The test results indicated that drought stress had the greatest impact on the crushing resistance of cucumber seedlings, and the crushing resistance was only 74.55% when the water content in the substrate was 50%, far lower than any other treatment. The reason was that the overall root system was not developed enough, and the substrate particles were relatively loose, which led to the poor seedling pot, so serious damage would happen as shown in Figure 5b,c. In addition, adding a proper proportion of biochar to the substrate could also effectively improve the stability of the substrate aggregate [46], thereby enhancing the stiffness of the substrate. Therefore,

sufficient water and a good rigidity of the substrate with a strong root system were the preconditions to ensure the crushing resistance of the seedling pots. The final results also showed that the crushing resistance of the seedling pots cultivated under the $B_5–W_{80}–N_{50}$ treatment was the best, but the difference between the $B_5–W_{65}–N_{50}$ and $B_5–W_{65}–N_{50}$ treatments was not obvious.

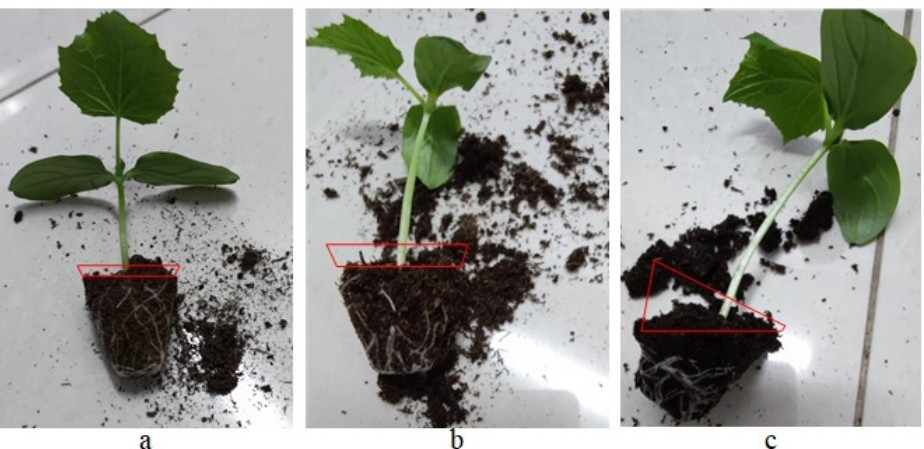

**Figure 5.** Effects of dropping damage with different treatments of cucumber seedlings: (**a**) seedling with less breakage; (**b**,**c**) Seedlings with more breakage. Note: The area marked in the red line is the damaged area after the seedling was dropped.

The analysis of the growth parameters and determination of leaf gas exchange parameters of cucumber seedlings after transplanting indicated that the seedlings with good growth before transplanting would grow more vigorously after transplanting. The addition of biochar in an appropriate proportion promoted the growth and the leaf surface area of cucumber seedlings, so the photosynthesis of cucumber seedlings was promoted with the increase in the total dry biomass of cucumber seedlings [45]. The value of stomatal conductance was a direct response of the seedling leaves to the water content; the stomata of the leaves would totally open when the water was sufficient, and with the increase in transpiration rate, the external $CO_2$ could enter smoothly from the stomata, greatly improving the photosynthetic rate [47]. When the water content of the substrate decreased, the seedlings were subject to drought stress, and some stomata in the leaves would close. Under this condition, the transpiration rate decreases, and external $CO_2$ is blocked from entering the leaves, leading to a certain reduction in the photosynthetic rate and increasing the consumption of intercellular $CO_2$ [48,49], ultimately leading to the increase in the intercellular $CO_2$ concentration of seedlings under the $W_{50}$ treatment. In addition, the cucumber seedlings under drought stress may have an autoimmune reaction and actively close the stomata to reduce the loss of transpiration, so as to mitigate the adverse effects of drought stress. Although the photosynthetic rate decreased, the water use efficiency was improved [50]. Compared with no application of N, the proper amount of N–fertilization increased the stomatal conductance and transpiration rate of leaves, thus significantly reducing the intercellular $CO_2$ concentration, and ultimately promoting the net photosynthetic rate of cucumber seedlings [51]. However, with the increase in N–fertilization, the amount of N needed by the seedlings was excessive, so the growth of the seedlings is inhibited, and the stomata of the leaves would largely close due to the damage, which decreases the stomatal conductance and transpiration rate, leading to a decrease in the photosynthetic rate [52]. Therefore, sufficient water content and proper N application are the premises to ensure the growth of cucumber seedlings.

## 5. Conclusions

This study demonstrated the significant and complex interactions among biochar rate, water content, and N–fertilization on the growth parameters, NUE, WUE, and crushing

resistance of cucumber seedlings before transplanting with the growth parameters and gas exchange parameters after transplanting. The growth parameters of cucumber seedlings decreased significantly with the increase in biochar rate, so it is necessary to control the proportion of biochar in the substrate between 5% and 10%. The growth parameters of cucumber seedlings significantly increased with the increase in water content, but the influence of water use efficiency was different, and a higher water content could alleviate the inhibition of high N on the growth of seedlings. Low N was still the best choice for the cultivation of seedlings, which increases water and N use efficiency. In addition, with a good water–holding capacity and adsorbing ability, biochar could significantly alleviate drought stress and the inhibition of high N on the growth of seedlings. The addition of biochar could effectively improve the strength of the substrate, but drought stress and high N lead to a poor root system with a loose substrate, so sufficient water and low N would have a significant interaction with biochar to promote the crushing resistance of seedlings when the biochar rate is 5%. The measurement of seedlings transplanted into the flowerpot indicated that the seedlings with superior growth before transplanting grow better after transplanting, with better photosynthesis. To sum up, seedlings obtained optimal growth parameters and crushing resistance when the biochar rate was 5%, water content was 80%, and N–fertilization was 50% in the substrate, meeting the requirements of mechanical transplanting, and indicating that these seedlings could also grow rapidly in the later period of transplanting.

**Supplementary Materials:** The following supporting information can be downloaded at: https://www.mdpi.com/article/10.3390/agronomy12112855/s1, Figure S1: The cultivation of the seedlings before transplanting; Figure S2: The measurement of plant height and total root length; Figure S3: The cultivation of the seedlings after transplanting; Table S1: The implication of 27 numbers and CK on the *x*–axis in the Figure.

**Author Contributions:** Conceptualization, G.M. and H.M.; methodology, G.M., H.M. and Y.L.; software, X.C.; validation, H.M., J.H. and L.H.; formal analysis, G.M., X.C. and Y.L.; investigation, X.C.; resources, Y.L.; data curation, G.M.; writing—original draft preparation, G.M.; writing—review and editing, G.M., H.M., J.H., Y.L. and X.C.; supervision, H.M., J.H. and L.H.; project administration, H.M., J.H. and L.H; funding acquisition, H.M. All authors have read and agreed to the published version of the manuscript.

**Funding:** This research was funded by a grant from the National Natural Science Foundation of China (51975258), the Priority Academic Program Development of Jiangsu Higher Education Institutions (PAPD–2018–87), the Demonstration and Application Project of Precise and Efficient Transplanting Equipment Industrialization (TC210H02X), the Open Fund of the Ministry of Education Key Laboratory of Modern Agricultural Equipment and Technology & High–tech Key Laboratory of Agricultural Equipment and Intelligence of Jiangsu Province (JNZ201910), and the Open Fund of Key Laboratory of Modern Agricultural Engineering of Tarim University (TDNG2020204).

**Institutional Review Board Statement:** Not applicable.

**Informed Consent Statement:** Not applicable.

**Data Availability Statement:** The data presented in this study are available from the corresponding author upon reasonable request.

**Acknowledgments:** The authors would like to thank the Key Laboratory of Agricultural Engineering at Jiangsu University for supporting the experimental conditions of the research.

**Conflicts of Interest:** The authors declare no conflict of interest.

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
