# Peer review of "Effects of Compound Biochar Substrate Coupled with Water and Nitrogen on the Growth of Cucumber Plug Seedlings"

_agronomy, doi:10.3390/agronomy12112855_

Round 1

Reviewer 1 Report

Comments for Author,

I have read your paper and enjoyed it. I think this paper is within the scope of the journal and well-designed. However, I have only minor comments before accepting this valuable paper.

The author mentioned in many places in the introduction regarding organic and liquid organic fertilizer, biochar, and its crucial role to reduce the detrimental effects of some environmental issue sources ın my opinion, this section is needed to support some refs, and improve, please mention these refs in the introduction regarding other organic and inorganic fertilizer sources, biochar application

https://doi.org/10.1038/s41598-019-51587-x.

Please incorporate all of them in the introduction.

Results and the used tables and figures reflect the paper, so,  it is fine. 

The discussion section is quite fine.

In my opinion, the conclusion section could be extended with added some information about effect of biochar. 

Best Regards

Author Response

The authors thank the reviewer 1 for affirming the content of our manuscript, and also thank his/her comprehensive comments for improving the quality of this manuscript.

In the revised manuscript, the words in blue colour were edited again according to your and other reviewers’ comments.

Point 1: The author mentioned in many places in the introduction regarding organic and liquid organic fertilizer, biochar, and its crucial role to reduce the detrimental effects of some environmental issue sources in my opinion, this section is needed to support some refs, and improve, please mention these refs in the introduction regarding other organic and inorganic fertilizer sources, biochar application https://doi.org/10.1038/s41598-019-51587-x.

Please incorporate all of them in the introduction.

Response 1:  According to your and other reviewers’ comments, we modified the description of biochar, and added some necessary references including ‘doi.org/10.1038/s41598-019-51587-x’ in line 65-79 (the new line number in the revised manuscript, the same below).

Point 2: In my opinion, the conclusion section could be extended with added some information about effect of biochar.

Response 2: Thank you for your comment, we added some information about the effect of biochar in conclusion, and the conclusion was also simplified according to other reviewers’ comments.

We gratefully acknowledge your comments and suggestions, which are very valuable in improving the quality of our manuscript.

Reviewer 2 Report

Abstract is well structured. All sections are well described and gave a good view of research to readers. But there is still some small points that need to be correct. 

Lines 71-77: Biochar is your main amendment but you didn't present enough background for biochar. Please add more references about its effects specially effect of biochar on soil water content and water availability for plant root. Because measuring water use efficiency is one of parts of your work. So, it's require to give readers a background in this matter. You can add these references to this section: 

  •  

10.1002/ldr.4006

Lines 132-135: ''Previous research indicated that 10% biochar mixed in the substrate could significantly promote the growth of plug seedlings [22], and this research refined the proportion of biochar on the base of 10% in order to find a more accurate biochar proportion added in the substrate which was more conducive to the growth of plug seedlings,...''. ----> You mentioned it one time in line 86. Don't need to repeat it again. Consider this work as a separate study with is stem on itself. So please delete it.

Lines 155-159:''The cucumber plug seedlings of each treatment would be cultivated in the incubator for 23 days (from 26 July to 18 August) until they were suitable for transplanting, and 8 seedlings would be selected from each treatment for testing before transplanting. The remaining 4 seedlings of each treatment would be transplanted into the flowerpot and cultivated until the flowering stage (from 18 August to 5 September) for subsequent tests.''.-----> Why the verbs' tense is future? Return whole this paragraph to past tense!

In Conclusion you repeated again some results and they made this section too long! Please make it short in one paragraph (max 15 lines).

Good luck!

Author Response

The authors thank the reviewer 2 for affirming the content of our manuscript, and also thank his/her comprehensive comments for improving the quality of this manuscript.

In the revised manuscript, the words in blue colour were edited again according to your and other reviewers’ comments.

Point 1: Lines 71-77: Biochar is your main amendment but you didn't present enough background for biochar. Please add more references about its effects specially effect of biochar on soil water content and water availability for plant root. Because measuring water use efficiency is one of parts of your work. So, it's require to give readers a background in this matter. You can add these references to this section: 10.1002/ldr.4006

Response 1:  According to your comment, we added some references including ‘doi:10.1002/ldr.4006’ about the effect of biochar on soil water content and water availability for plant root, and part of the description of biochar in the introduction was modified too in line 65-79 (the new line number in the revised manuscript, the same below).

Point 2: Lines 132-135: ''Previous research indicated that 10% biochar mixed in the substrate could significantly promote the growth of plug seedlings [22], and this research refined the proportion of biochar on the base of 10% in order to find a more accurate biochar proportion added in the substrate which was more conducive to the growth of plug seedlings,...''. ----> You mentioned it one time in line 86. Don't need to repeat it again. Consider this work as a separate study with is stem on itself. So please delete it.

Response 2: According to your comment, we deleted ‘Previous research indicated that 10% biochar mixed in the substrate could significantly promote the growth of plug seedlings [22], and this research refined the proportion of biochar on the base of 10% in order to find a more accurate biochar proportion added in the substrate which was more conducive to the growth of plug seedlings’ in original line 132-135, and changed the sentence into ‘Based on previous research, the three biochar rates included 5%, 10% and 15%, which was represented as B5, B10 and B15 respectively.’ in line 139-140.

Point 3: Lines 155-159:''The cucumber plug seedlings of each treatment would be cultivated in the incubator for 23 days (from 26 July to 18 August) until they were suitable for transplanting, and 8 seedlings would be selected from each treatment for testing before transplanting. The remaining 4 seedlings of each treatment would be transplanted into the flowerpot and cultivated until the flowering stage (from 18 August to 5 September) for subsequent tests.''.-----> Why the verbs' tense is future? Return whole this paragraph to past tense!

Response 3: Thank you for your comment, we returned whole this paragraph to past tense in line 156-160.

Point 4: In Conclusion you repeated again some results and they made this section too long! Please make it short in one paragraph (max 15 lines).

Response 4: Thank you for your comment, we deleted some repeated results in conclusion. Since the left side of the text in manuscript was indented 4.2cm, we tried our best to simplify the conclusion into one paragraph with 21 lines, 286 words in total.

We gratefully acknowledge your comments and suggestions, which are very valuable in improving the quality of our manuscript.

Reviewer 3 Report

Agronomy: Effects of compound biochar substrate coupled with water and nitrogen on the growth of cucumber plug seedlings

This paper describes investigations into the effects of biochar, water, and N-fertilizer on growth of cucumber seedlings.  I think the Results would be of interest to cucumber growers and provide valuable data with respect to the use of biochar in horticultural settings.

However, the results are presented in quite an unconventional way – tables being used to summarise the effects of the main factors (water, N, bochar) and then figures used to display the various two-way and three-way combinations. Regrettably, this is not done well, and the authors need to find a better way of displaying their results; maybe some form of heat map approach is required, or at least better use of faceting multiple graphs in one figure?

Also, the English is of quite a low standard, and the paper is actually quite difficult to read in places, and a major review of the text is required before this paper should be considered for publication.

Some minor comments below

Abstract

ln 18 & 20  what are DM and WUE – not defined yet

ln 23 indices not indexes

ln 24 delete totally

Introduction

ln 34 delete first sentence

ln 40 delete great

The introduction is not written well and requires improvement

Methods

ln 113 please sort out the grammar in this sentence
ln 117 “The substrate was composed of peat and biochar.” but ln 121 “The substrate was composed of peat and soil in 1:1 ratio”.   which is it?

Table 1 not needed – methods are already in text

ln 122 ‘certain portion’ – what actually was the portion?
ln 231 data were
ln 235 delete these last four lines.  not needed
ln 138 I think you need to include the date after the author when making statements like this.

Results

This is Table 2?  Sort out Table numbers.
Are the units correct – the plant heights are only around 50mm whereas the roots are over 1m in length.   Also, use same decimal places in each column.

Same with Table 4 – is the root length of a cucumber really 11m ?  Is this total root length ?  Need to make this clear.

Fig 1 does not show the combination of all three treatments – you need to find a way of illustrating all 27 treatment combinations. Fig 1d and Fig 2d and Fig 3d and Fig 4d on x-axis just gives 27 numbers – this makes it impossible to tell which treatment is which.

Discussion

The Discussion is too long and should focus more on the results obtained in this study and their implications for growing cucumbers.

Author Response

The authors thank the reviewer 3 for affirming the content of our manuscript, and also thank his/her comprehensive comments for improving the quality of this manuscript.

We mainly analysed the combination of three factors through Tables in the manuscript. Since there were many parameters measured in this research, considering the length of the paper, four main parameters were shown in Fig 1, Fig 2, Fig 3, and Fig 4 as a supplement to the Tables for better intuitive display of the combination of three factors. Fig a, Fig b and Fig c in those Figures showed the two-way combination between Biochar-Water, Biochar-N and Water-N respectively. Fig d in those Figures showed the final result of the three-way combination among three factors, which was used to display the comparation of 27 treatments and CK treatment in order to obtain the best treatment for seedling cultivation. We are very grateful for your advice, which makes us realize that there may be a better way to represent the combination of factors, which will make great help to our future research work. Besides, we asked one of our native English-speaking colleagues to improve the English standard for us.

In the revised manuscript, the words in blue colour were edited again according to your and other reviewers’ comments.

Abstract

Point 1: ln 18 & 20  what are DM and WUE – not defined yet

Response 1:  According to your comment, we changed ‘Total DM’ into ‘Total Dry Matter’ in line 20, and changed ‘WUE’ into ‘Water use efficiency’ in line 18 & 20 (the new line number in the revised manuscript, the same below).

Point 2: ln 23 indices not indexes

Response 2: According to your comment, we changed ‘indexes’ into ‘indices’ in line 24.

Point 3: ln 24 delete totally

Response 3: According to your comment, we deleted ‘totally’ in line 25.

Introduction

Point 4: ln 34 delete first sentence

Response 4: According to your comment, we deleted first sentence in original line 34.

Point 5: ln 40 delete great

Response 5: According to your comment, we deleted ‘great’ in line 39.

Point 6: The introduction is not written well and requires improvement

Response 6: We deleted some references in the introduction, and some necessary modifications with some references was added in the introduction (especially the biochar part) were made according to your comment. Besides, we revised the introduction according to the comments of other reviewers.

Methods

Point 7: ln 113 please sort out the grammar in this sentence

Response 7: According to your comment, we sorted out the grammar, and changed the sentence into ‘The cucumber seedlings tested in this study were “Jinyou 1”, which were provided by cucumber research institute in Tianjin Academy of Agricultural Sciences, Tianjin, China.’ in line 119-121.

Point 8: ln 117 “The substrate was composed of peat and biochar.” but ln 121 “The substrate was composed of peat and soil in 1:1 ratio”.   which is it?

Response 8: We are sorry to confuse you about that, so we changed ‘The substrate was composed of peat and biochar’ into ‘The substrate used before transplanting was composed of peat and biochar’ in line 123-124, and ‘The substrate was composed of peat and soil in 1:1 ratio’ into ‘The substrate used after transplanting was composed of peat and soil in 1:1 ratio’ in line 127-128.

Point 9: Table 1 not needed – methods are already in text

Response 9: According to your comment, we deleted original Table 1 to avoid duplication.

Point 10: ln 122 ‘certain portion’ – what actually was the portion?

Response 10: We are sorry that we did not explain ‘certain portion’ clearly in original manuscript. Since the nutrient solution is prepared according to Table 1 in the reference, so we changed ‘a certain proportion of nutrient solution’ into ‘The nutrient solution prepared according to reference’ in line 128-129.

Point 11: ln 231 data were

Response 11: According to your comment, we changed ‘data was’ into ‘data were’ in line 232.

Point 12: ln 235 delete these last four lines.  not needed

Response 12: According to your comment, we deleted last four lines about explanation of P value in manuscript, and the new sentence was shown in line 232-235.

Point 13: ln 138 I think you need to include the date after the author when making statements like this.

Response 13: According to your comment, we added the date after the author in line 141, and the same as ‘Yang (1991)’ in line 50, ‘Min (2016)’ in line 57, ‘Ting (1990)’ in line 58, ‘Qu (2018)’ in line 60.

Results

Point 14: This is Table 2?  Sort out Table numbers.

Response14: We are sorry that we made a mistake about the order of the Tables in original manuscript. According to your comment, we deleted original Table 1 and reordered the Tables.

Point 15: Are the units correct – the plant heights are only around 50mm whereas the roots are over 1m in length. Also, use same decimal places in each column.

Response 15: Thank you for your comment. After washing tidy with tap water, the root parameters of seedlings were direct measured by the Perfection V700 photo scanner and analysed by the WinRHIZO root analysis software, so the total length of root was obtained from the software. The root system of plug seedlings was dense, and the total length was the sum of all root lengths, so these data were accurate and we changed ‘root length’ into ‘total root length’ in new manuscript for better understanding. Besides, the decimal places in each column were modified to the same according to your comment.

Point 16: Same with Table 4 – is the root length of a cucumber really 11m ?  Is this total root length ?  Need to make this clear.

Response 16: The root determination method was the same as response 15. Since the root parameters were measured from the seedlings which cultivated in the plug tray for 23 days, and then cultivated in a bigger flowerpot for about 15 days, the roots became more dispersed and longer in the substrate, and the cleaned root system of each seedling must be divided into three parts for measurement which was limited by the size of the machine root test area, so the root length was accurate. Besides, we changed ‘root length’ into ‘total root length’ in new manuscript for better understanding.

Point 17: Fig 1 does not show the combination of all three treatments – you need to find a way of illustrating all 27 treatment combinations. Fig 1d and Fig 2d and Fig 3d and Fig 4d on x-axis just gives 27 numbers – this makes it impossible to tell which treatment is which.

Response 17: Thank you for your precious comments. Since the results of all treatments were analysed by the SPSS software and shown in Table 1 and Table 2, we mainly analysed the combination of three factors through these Tables. As a supplement to the Tables, some main parameters were selected to draw as image (Fig 1, Fig 2, Fig 3, and Fig 4) for better intuitive display of the combination of three factors. Fig a, Fig b and Fig c in the Figures showed the combination between Biochar-Water, Biochar-N and Water-N respectively, while Fig d in the Figures showed the final result of the three-way combination among three factors, which was used to display the comparation results of 27 treatments and CK treatment in order to obtain the best treatment for seedling cultivation. Besides, because the layout of the picture was not enough, we used different numbers to represent each treatment, and the implication of 27 numbers on x-axis in Fig 1d, Fig 2d, Fig 3d and Fig 4d were added in the supplementary materials which was marked as Supplementary Table S1, and it was explained in line 280-281 of the manuscript.

Discussion

Point 18: The Discussion is too long and should focus more on the results obtained in this study and their implications for growing cucumbers.

Response 18: According to your comment, we revised the discussion. The original discussion was analysed according to each measured parameters of the cucumber seedlings, which led to a long discussion. We reorganized and mainly analysed the discussion from four aspects: the growth parameters of cucumber seedlings before transplanting, the water and N use efficiency of cucumber seedlings before transplanting, the crushing resistance of cucumber seedlings before transplanting and the growth parameters of cucumber seedlings after transplanting. Besides, we deleted and merged some duplicate descriptions, and tried our best to simplified each aspect in the discussion, while some necessary contents were also added in this part.

We gratefully acknowledge your comments and suggestions, which are very valuable in improving the quality of our manuscript.

Round 2

Reviewer 3 Report

The authors have addressed the comments and suggestions raised by the reviewers and this version of the manuscript is much improved from the first submission. 

There are still some minor problems with the English: for example, there are many instances where the 'future tense' is used ('would be) where the past tense is more appropriate (was or were). Some of these issues might be corrected at the proof stages.